# Comparison of the Modulating Effect of Anthocyanin-Rich Sour Cherry Extract on Occludin and ZO-1 on Caco-2 and HUVEC Cultures

**DOI:** 10.3390/ijms23169036

**Published:** 2022-08-12

**Authors:** Judit Remenyik, Attila Biró, Ágnes Klusóczki, Krisztián Zoltán Juhász, Tímea Szendi-Szatmári, Ádám Kenesei, Erzsébet Szőllősi, Gábor Vasvári, László Stündl, Ferenc Fenyvesi, Judit Váradi, Arnold Markovics

**Affiliations:** 1Institute of Food Technology, Faculty of Agricultural and Food Sciences and Environmental Management, University of Debrecen, H-4032 Debrecen, Hungary; 2Institute of Healthcare Industry, University of Debrecen, H-4032 Debrecen, Hungary; 3Department of Biophysics and Cell Biology, Faculty of Medicine, University of Debrecen, H-4032 Debrecen, Hungary; 4Department of Pharmaceutical Technology, Faculty of Pharmacy, University of Debrecen, H-4032 Debrecen, Hungary

**Keywords:** tight junction (TJ), zonula occludens-1 (ZO-1), occludin, Caco-2, HUVEC

## Abstract

Increased permeability of the epithelial and endothelial cell layers results in the onset of pathogenic mechanisms. In both cell types, cell–cell connections play a regulatory role in altering membrane permeability. The aim of this study was to investigate the modulating effect of anthocyanin-rich extract (AC) on TJ proteins in inflammatory Caco-2 and HUVEC monolayers. Distribution of Occludin and zonula occludens-1 (ZO-1) were investigated by immunohistochemical staining and the protein levels were measured by flow cytometry. The mRNA expression was determined by quantitative real-time PCR. The transepithelial electrical resistance (TEER) values were measured during a permeability assay on HUVEC cell culture. As a result of inflammatory induction by TNF-α, redistribution of proteins was observed in Caco-2 cell culture, which was reduced by AC treatment. In HUVEC cell culture, the decrease in protein and mRNA expression was more dominant during inflammatory induction, which was compensated for by the AC treatment. Overall, AC positively affected the expression of the examined cell-binding structures forming the membrane on both cell types.

## 1. Introduction

The intestinal epithelium represents a protective physical barrier and actively contributes to the mucosal immune system [1]. Vascular endothelial cells form a continuous monolayer, which constitutes a dynamic and highly effective cellular barrier between the vessel wall and bloodstream; they are also actively involved in the functioning of the immune system [2]. Epithelial cells and endothelial cells are directly adjacent, indicating active inter-barrier communication. These cells are associated with each other via intercellular junctions that differ in their morphological appearance, composition, and function. One manifestation of connectivity is the appearance of tight junctions (TJs) on the cell membrane [3,4].

TJs are intercellular permeability seals, which regulate diffusion of particles with size and charge selectivity, and thereby play a crucial role in determining paracellular permeability. Epithelial and endothelial TJs are the key regulators of paracellular transport of macromolecules [5]. Many molecular components of tight junctions have now been identified [6]. Occludin is one type of transmembrane domain-containing protein, which is incorporated into, or localized very close to, the claudin base tight junction strands. Occludin and claudins are linked to a number of cytoplasmic scaffolding and regulatory proteins ZO-1, ZO-2, ZO-3 and cingulin [7]. 

TJs plays a crucial role as a physical and functional barrier against the paracellular penetration of dangerous substances present in the lumen (including bacteria, bacterial toxins, digestive enzymes, degraded food products, bacterial byproducts) [8,9,10,11]. Epithelial TJs maintain the intestinal barrier while regulating permeability of nutrients, ions, and water. Endothelial TJs have a key role in vascular permeability and leukocyte extravasation [12]. The modification of function and paracellular permeability of the TJ barrier results in barrier disruption of TJs and increased paracellular permeability, leading to sustained inflammation and tissue damage [5]. Furthermore, polarized basolateral intestinal secretion of inflammatory mediators, followed by activation of different inflammatory pathways in endothelial cells, effectively induces neutrophil extravasation from the vascular system, thereby contributing to the maintenance of intestinal inflammation [2,13]. 

The cytokine network plays a very important role in the regulation of inflammation, of which tumor necrosis factor-α (TNF-α) is a key player. The importance of TNF-α has been highlighted for the last decade, through both experiments on mouse models of intestinal inflammation and pivotal human therapeutic trials with the chimeric monoclonal antibody to TNF-α: Infliximab [14,15]. TNF-α has been shown to increase permeability in several epithelial models. Studies on human colon carcinoma cell line (Caco-2) have shown that the effects of TNF-α on altered tight junction permeability are regulated through NF-κB activation [16,17]. It has been stated that TNF-α regulates intestinal permeability by modulating signaling pathways, mainly NF-κB, affecting the structure and functionality of the tight junctions [18,19,20]. In our previous study, the activation of the NF-κB pathway was confirmed in this context [21]. 

There is evidence that TJs rearrange and modify their transcription in epithelial and endothelial cells due to inflammation, resulting in increased paracellular permeability [22,23]. However, it has been shown, that many natural substances can be able to moderate the TJ rearrangements and decreasing in permeability due to inflammation [24,25].

In our previous studies, we demonstrated the beneficial effects of anthocyanins isolated from Hungarian sour cherries, in both in vitro and in vivo investigations [26,27,28,29,30,31,32]. In this study we show that the anthocyanin-rich sour cherry extract (AC) exerts a pleiotropic effect, including anti-oxidative, anti-inflammatory, hemostatic and vasoactive effects, indicating that this substance could be protective in inflammatory bowel diseases and inflammatory endothelial dysfunction. Anthocyanins are considered one of the flavonoids; however, they have a positive charge at the oxygen atom of the C-ring of anthocyanidin structure, it is called flavylium ion. The colored pigments of anthocyanin, from blackcurrants, berries, and other types of red or blue fruits, have strong antioxidant effects. The most common anthocyanidins are cyanidin, delphinidin, malvidin, petunidin, peonidin and pelargonidin, the glycosylated form of each being distributed in different percentages in each plant [33,34]. 

Previously, we induced inflammation using a combination of 25 ng/mL IL-1β and 50 ng/mL TNF-α and we applied 50 µM AC for 24 h to investigate the barrier function of the Caco-2 cell monolayer. We observed that this induction cocktail caused significant IL-6 and IL-8 release into apical compartments; however, AC pre-treatment led to reduced levels of the abovementioned cytokines. In permeability and NF-κB-related experiments, we found that 50 µM AC extract inhibited the effects of cytokine-induction [21]. However, we did not investigate the barrier-modulation effect of AC. 

Human umbilical vein endothelial cell (HUVEC) and Caco-2 cultures are widely used barrier models for the investigation of inflammatory processes in various chronic diseases. The effects of anthocyanins on barrier function have been the subject of intensive research. In our study, we focused on Occludin and ZO-1 which are the main components of TJ strands in Caco-2 and HUVEC. The proteins expressions were investigated using immunofluorescence and their mRNA expressions were determined by quantitative real-time PCR. The TEER was measured, and apparent permeability coefficients (Papp) were determined in HUVEC and compared to our previous result on Caco-2 cell culture. Knowing the above, the barrier-modulation effect of anthocyanin-rich sour cherry extract (AC) was investigated, using an in vitro model system of TNF-α induced inflammation in endothelial and Caco-2 cells. 

## 2. Results

### 2.1. Real-Time Monitoring of Cell Index (RTCA) in HUVEC

In our previous studies, a non-cytotoxic concentration of AC in HUVEC was determined using MTT-assay and fluorescent labelling with: DilC_1_(5) (examination of apoptosis); and SYTOX Green (investigation of necrosis) dyes. These tests showed that up to 170 μM AC does not show significant cytotoxic effects [31]. In addition, similar to Caco-2 cells, three different concentrations of AC were also checked by RTCA method on the HUVEC monolayer. After the cell layer reached the maximum cell index (CI), wells were treated with different concentrations of AC. Compared to the control, the 85 and 170 μM concentrations of AC treatment did not result in a decrease in CI, rather, a slight increase (Figure 1). Nevertheless, a reversible decrease in the CI curve was observed with the treatment of 850 μM AC concentration. Thus, the two lower AC concentrations did not cause cytotoxicity in HUVEC and were found to be safe to use. Therefore, treatment with a concentration of 85 μM was applied in our further studies.

### 2.2. Permeability Assay on HUVECs

HUVECs were seeded and grown on Corning Transwell^®^ polycarbonate filters. TEER values were checked from day 14; values did not change significantly in the third week and remained between 250 and 450 Ω*cm^2^. The values of the control P_app_ on the HUVECs were of the order of 10^−5^, which is one order of magnitude higher than the values determined on the Caco-2 in our previous study [21]. In the case of TNF-α induction, P_app_ values did not change significantly, as in the combination-treated (AC + TNF-α) samples (see Figure 2), compared to the control.

### 2.3. Immunofluorescence

#### 2.3.1. Immunohistochemical Staining of Occludin and ZO-1

On untreated Caco-2 cells, Occludin showed a peripheral arrangement, as did ZO-1. However, with an inflammatory inducer (TNF-α), the arrangement of Occludin and ZO-1 became characteristic zigzag patterns, compared to the control. In the presence of inflammatory inducer, AC treatment was able to moderate the development of the zigzag pattern. In the case of Occludin, the combined treatment also resulted in a stronger, thicker pattern (Figure 3).

Immunohistochemical staining of TJ proteins in HUVEC cells showed significantly different, central plasma localization, compared to Caco-2 cells. ZO-1 expression was significantly decreased after TNF-α treatment in HUVECs (Figure 4).

In contrast, the fluorescent intensity of Alexa Fluor 647-labeled Occludin was significantly increased, compared to the control. AC treatment was able to compensate for this change in the cause of inflammation, to a significant extent, for both markers. 

#### 2.3.2. Immunofluorescence Determination of Occludin and ZO-1 Proteins by Flow Cytometry

The levels of TJ proteins Occludin and ZO-1 in Caco-2 cell cultures were determined in the absence or presence of AC and TNF-α, by flow cytometry. The presence of TNF-α significantly decreased the expression of ZO-1. A similar tendency could be observed in the presence of AC in the culture media. A combination treatment of TNF-α and AC led to a lower expression level, compared to the control cells; however, the highest expression level of ZO-1 could be measured in the presence of inflammatory cytokine and AC combined, in contrast with TNF-α-treated cells, which showed the lowest level of expression (Figure 5A).

Examination of Occludin expression by flow cytometry revealed that no significant changes were shown as a result of TNF-α or AC alone, although significant Occludin expression could be observed due to AC + TNF-α treatment, compared to TNF-α treatment alone (Figure 5B).

Similar to Caco-2 cells, expressions of Occludin and ZO-1 proteins were evaluated in HUVECs in the absence or presence of AC and TNF-α by flow cytometry. Interestingly, TNF-α treatment significantly reduced the expression levels of both markers, in contrast to the results of microscopic studies. Nevertheless, as in the case of the Caco-2 cells, this decrease was significantly positively influenced by the combined treatment of TNF-α and AC in HUVECs (Figure 6).

### 2.4. Quantitative Real-Time PCR (qPCR)

Expression values of ZO-1 and Occludin genes were measured in untreated (C), inflammatory cytokine-treated (TNF-α) and AC + TNF-α-treated Caco-2 cells. Real-time quantitative PCR showed a significantly reduced level of ZO-1 mRNA induced by TNF-α. The AC + TNF-α combination cocktail resulted in significantly increased ZO-1 mRNA expression, compared to TNF-α induced cells. (Figure 7A). The mRNA level of Occludin significantly decreased after induction with TNF-α. However, we could observe upregulation of Occludin gene after 24 h combination treatment of AC and TNF-α, compared to TNF-α treated cells. (Figure 7B).

The genes of ZO-1 and Occludin were assessed by qPCR in HUVECs. As we expected, the mRNA levels of the investigated tight junctions were significantly reduced in the presence of TNF-α. AC treatment was able to elevate the observed decrease, in both markers, to the level of control (Figure 8A,B). Based on our observations, the changes in the expression of ZO-1 and Occludin genes observed in HUVECs are consistent with the flow cytometry results of the protein assays.

## 3. Discussion

In our previous study, we investigated the protective effects of the pure sour cherry anthocyanin extract under inflammation conditions on HUVEC and Caco-2 cell cultures [21]. We tested the cytotoxicity effect of AC by RTCA and verified that it has no toxicity effect on Caco-2 cell culture in the applied concentration range. We found no significant concentration dependence (500, 50, 5, 0.5 µM) of AC, however the highest concentration (500 µM) of AC treatment led to a significantly decreased cell index, compared to the control curve. Although this reduction was even less than the positive control, we demonstrated the safety of application of AC in the abovementioned concentrations. 

The endothelial cells are more sensitive to different stimuli (cytokine, neurotransmitters, toxins, hypoxia) because these modify the organization of TJ proteins. The disruption of the endothelial barrier causes leakage, and this is reflected in the decrease in TEER and CI [4,35]. Thus, the same concentration cannot be used on Caco-2 and HUVEC cultures. In our previous studies, we showed that AC extract, up to 100 μg/mL, does not show significant cytotoxic effects [36]. In our current study we confirmed these results by using real time cell analysis (RTCA). At the higher concentration (850 µM), the extract temporarily destabilized the monolayer, so it was not used in tests. In our further studies, HUVEC cells were treated with 85 μM AC.

The TEER value, measured on the confluent monolayer, and the permeability studies are important indicators of the membrane barrier function [37,38]. Thus, the analysis of these parameters, in addition to the study of TJ proteins, is unavoidable. Studies examining the barrier function of Caco-2 and HUVEC monolayers confirm the close relationship between these two parameters and Occludin expression [39]. In our previous study, we reported the protective effects of AC on membrane permeability, in an inflamed Caco-2 monolayer [21]. The TEER values were measured 24 h after induction with 25 ng/mL IL-1 and 50 ng/mL TNF-α in a basolateral compartment. Compared to the untreated control, significantly decreased TEER values were measured in cytokine treated wells, however with AC pre-treated samples, the AC moderated the cytokine-induced TEER value decrease. In the present study, paracellular transport was investigated in HUVECs. Development of the HUVEC monolayer was controlled by measuring the TEER 10 days after the seeding of cells. However, checking the TEER at 2 days showed that the values had not significantly increased. The TEER values increased between 250 and 450 on day 21 and then the permeability assay was performed. Figure 2 shows the P_app_ values; there were no significant differences in the permeability values of the different groups, compared to the untreated control. However, control P_app_ values in the HUVECs were one order of magnitude higher than previous P_app_ control values in Caco-2, possibly indicating a difference in the strength of the two monolayers’ closing relationships.

Following preliminary experiments, immunohistochemical staining was performed, and different localizations of TJ proteins were investigated on Caco-2 and HUVEC cells. In the control group of Caco-2 culture, cell–cell attachments in the monolayers were continuous, without gaps. In cytokine (TNF-α) treated cells, the pattern of the staining changed: intercellular gaps, fragmented junctional staining and cytoplasmic redistribution of junctional proteins were observed. The immunostaining pattern of TJ proteins in Caco-2 cells treated with AC + TNF-α was similar to that seen in the control group: zipper-like irregularities were reduced in the presence of AC. In the Caco-2 monolayer, we found mainly morphological features, supporting the previous studies’ findings that the TJ and AJ proteins of epithelial cells show a higher order than those observed in endothelial cells [35,40]. 

One of the transmembrane representatives of TJ proteins that appears closer to the apical surface of cells is Occludin [41]. Occludin plays an important role in the regulation of paracellular transport and in the formation of cell–cell connections in both epithelial and endothelial cells. In immunohistochemical staining, a definite pattern around Caco-2 cells indicates this dominance, which is less apparent in HUVEC cells [40]. This may be due to the different origins of the cells, as the Caco-2 cell line is immortalized and the HUVEC is a primary isolated cell culture. TJ proteins are dynamically organized according to the proliferation rate of the cell type, thus a different staining pattern was observed in the different cell cultures [42]. 

For the induction of inflammation in the case of Occludin, the zipper-like pattern on Caco-2 cells was reduced with AC treatment. This may indicate the effective intervention of AC in the phosphorylation-dephosphorylation process of Occludin [43]. 

Nevertheless, the staining intensity of Occludin and the change in cytometric Occludin concentrations are much more informative in HUVEC cell culture. TNF-α, as an inflammatory inducer, decreases Occludin concentration, which was confirmed by lighter immunostaining. A similar trend was observed in the study in which HUVECs were induced by LPS and the presence of Occludin was examined by Western blot [39].

In the case of HUVEC, the localization of ZO-1 is not limited to the cell membrane but is scattered in the cytoplasm. TNF-α treatment decreased the amount of Occludin expressed in the culture, indicating an increase in endothelial monolayer fenestration. This phenomenon was significantly reduced by AC treatment. 

In summary, endothelial and intestinal epithelial TJs play an essential role in vascular and intestinal homeostasis and disease, and targeting TJ-related biochemical and signaling pathways by AC might be a new therapeutic strategy for the treatment of a broad spectrum of human diseases in the near future. However, of course, further in vivo and human studies are needed, regarding the medical utility of AC.

## 4. Materials and Methods

All chemicals were from Sigma-Aldrich (Budapest, Hungary) unless stated otherwise. Pure sour cherry anthocyanin extract was provided by the Department of Feed and Food Biotechnology, University of Debrecen (Hungary), as described previously (Homoki JR et al., 2016). 

### 4.1. Cell Culture

Human White colon adenocarcinoma (Caco-2) cells (European Collection of Cell Cultures (ECACC, UK) were isolated from a primary colonic tumor in a 72-year-old White male using the explant culture technique. During routine subculture, cells were grown in Dulbecco’s modified Eagle’s medium (DMEM) with the supplementation of 10% fetal bovine serum (FBS), 1% non-essential amino acid and penicillin–streptomycin solution, at 37 °C, in an incubator containing 5% CO_2_. The passage number of Caco-2 cells was 25–40 in the study.

HUVEC/TERT 2 was obtained from ATCC (ATCC, Manassas, VA, USA). The cell line was isolated from the vascular endothelium of a White female patient. Cells were maintained in M199 with the supplementation of 10% heat-inactivated FBS, 1% penicillin/streptomycin, 1% amphotericin B, 2 mM glutamine (Biosera, Nuaille, France), and Endothelial Cell Growth Medium-2 (Lonza, Basel, Switzerland), at 37 °C, in a Galaxy 170R incubator under 5% CO2 (Eppendorf, Hamburg, Germany). Adhesion of the cells was support with a 0.1% gelatin solution.

### 4.2. Permeability Assay on HUVECs

HUVECs were seeded at a density of 200,000 cells/well on Corning Transwell^®^ polycarbonate filters. Culture medium was replaced with fresh medium every two or three days in the Transwell^®^ inserts. TEER value of samples were checked from day 14, but the values were not significantly increased in the third week. Thus, the assay was performed at 21 days. Confluent and differentiated cell layers were pre-treated in the upper compartment, with 85 µM anthocyanin extract dissolved in cell culture medium, for 24 h. A total of 10 ng/mL TNF-α was added into the medium and the plates were incubated at 37 °C in an incubator with 5% CO_2_ for 20 h. HUVEC monolayers were washed and pre-incubated with Hanks’ balanced salt solution (HBSS) for 20 min, at 37 °C, and then incubated with 50 µg/mL Lucifer yellow (LY) dissolved in HBSS, in the upper compartment. Samples (100 µL) were collected from the lower compartment at 60, 90 and 120 min and the volume was supplemented with HBSS. Concentration of Lucifer yellow was determined by a fluorescence multiwell plate reader (Fluostar Optima, BMG Labtechnologies, Germany) at 450 nm excitation and 520 nm emission wavelength. The apparent permeability coefficients (P_app_) were calculated by the following equation:(1)$$P_{\mathrm{app}}\;\;=\frac{dQ}{dt}\cdot \frac{1}{\left(C_{0}\cdot A\right)}$$P_app_: apparent permeability coefficient (cm/s); *dQ/dt*: permeability rate of substances (mol/s); *C*_0_: initial concentration of the substances in the upper compartment (mol/mL); *A*: surface area of membrane (cm^2^).

### 4.3. Immunofluorescence

For the differentiation experiments, cells were seeded onto sterile microscope slides at a density of 50,000 cells per slide. The investigation was designed as follows: Caco-2 and HUVEC cells were divided into four groups, in accordance with the stimulants. The first group of slide cultures was un-treated and served as a negative control. The second group of cells was stimulated with 50 ng/mL tumor necrosis factor-alpha (TNF-α), which is a pro-inflammatory cytokine, for 24 h. The third group was pretreated with TNF-α in combination with anthocyanin (AC). AC was administered, at 100 µM concentration, to the cells for 24 h. 

Morphological changes were monitored by immunostaining for zonula occludens protein-1 (ZO-1) and Occludin plasma-membrane protein. After AC + TNF-α treatments, cell layers were washed with PBS and fixed with 1:1 mixture of cold methanol and ac-etone for 10 min. After a washing step, cells were blocked with 3% bovine serum albumin in PBS and incubated for 30 min. 

In the Caco-2, the next step was to add the primary antibodies: rabbit anti-ZO-1 and rabbit Occludin to the cells. Incubation, with Alexa Fluor-488 anti-rabbit secondary antibodies (Life Technologies, Invitrogen, Waltham, MA, USA), lasted for 1 h. Bisbenzimide dye (Hoechst 33342, Sigma-Aldrich, St. Louis, MO, USA) was used to stain cell nuclei. The sample staining was visualized by a Zeiss Axio Scope, A1 fluorescent microscope (HBO 100 lamp) (Carl Zeiss Microimaging GmhH, Göttingen, Germany). Images were analyzed with ZEN 2012 v.1.1.0.0 software (Carl Zeiss Microscopy GmbH, Göttingen, Germany). 

In both HUVEC and Caco-2, for the next step, primary labeled anti-ZO-1, tagged with FITC, and Alexa Fluor 647-labeled Occludin primary antibodies were added to the cells for 1 h. DAPI (4′,6-diamidino-2-phenylindole) was used to stain the HUVEC cells’ nuclei. The sample labeling was visualized by a Zeiss fluorescent microscope (Zeiss LSM 880 confocal microscope Göttingen, Germany). Images were analyzed with ZEN 2012 v.1.1.0.0 software (Carl Zeiss Microscopy GmbH, Göttingen, Germany). The staining intensity was measured using the mean gray value corrected for background, with imageJ FIJI 1.53c software.

### 4.4. Flow Cytometry

Flow cytometry experiments were used to quantify the expression of ZO-1 and Occludin. For these experiments, both cell types were trypsinized with 0.05% trypsin-EDTA solution, washed twice with Hank’s balanced salt solution (HBSS) and suspended in a cell concentration of 1 × 10^6^ cells/mL. Cells were fixed in 3.7% formaldehyde in HBSS. After a washing step, cells were permeabilized with 0.1% Tween-20-containing Hanks balanced salt solution, as a permeabilization buffer, for 30 min at room temperature (22–25 °C). After this incubation, cells were incubated with 10% FBS for at least 30 min at 37 °C to block the nonspecific binding sites. Then cells were incubated with 2 µg/mL primary anti-ZO-1/Occludin antibody for 1 h at 37 °C.

Cells were washed twice with HBSS and incubated with 5 µg/mL secondary antibody for 1 h at 37 °C in dark place. After incubation, cells were washed three times and divided into a 96-well plate. Cells were analyzed by Guava Easy Cyte 6HT-2L flow cytometer (Merck Ltd., Darmstadt, Germany) in four independent experiments.

### 4.5. Real-Time Monitoring of Cell Index (RTCA)

Kinetics of epithelial cell reaction to AC + TNF-α treatment was monitored by impedance measurement with real time cell analyser (RTCA), XCelligence system. RTCA presents a non-invasive, real time and label-free method which linearly correlates with growth, adherence and viability of cells. For background measurements, 100 µL cell culture medium was added to the wells, then cells were seeded—at a density of 2 × 10^4^ cells/well—into the coated 16-well plates, with integrated gold electrodes. Before seeding of the HUVECs, wells were coated with collagen. Then HUVECs were cultured for 6 days in a CO_2_ incubator, at 37 °C, and monitored every 30 min until the end of the experiment. When the CI reached the maximum value in each well, cells were treated with 85, 170, or 850 µM AC, except for control wells, in which only the medium was renewed.

### 4.6. qPCR

The messenger RNA (mRNA) expressions (ZO-1, Occludin), in response to the treatment with AC + TNF-α, were determined by quantitative PCR experiments. Caco-2 cells were pretreated with AC + TNF-α for 24 h before the experiments. Total RNA was isolated by Extrazol (BLIRT, Pomorskie, Poland). cDNA was reverse transcribed from 1 µg of total RNA using LunaScript RT SuperMix Kit (PCR Biosystems, London, UK). The reaction was implemented using the Luna Universal Probe qPCR Master Mix (PCR Biosystems, London, UK) with glyceraldehyde-3-phosphate dehydrogenase as an internal control. Q-PCR was performed on a Roche LightCycler 480 System (Roche, Basel, Switzerland) using the 5′ nuclease assay. The results were expressed relative to 100% of the control group.

## Figures and Tables

**Figure 1 ijms-23-09036-f001:**
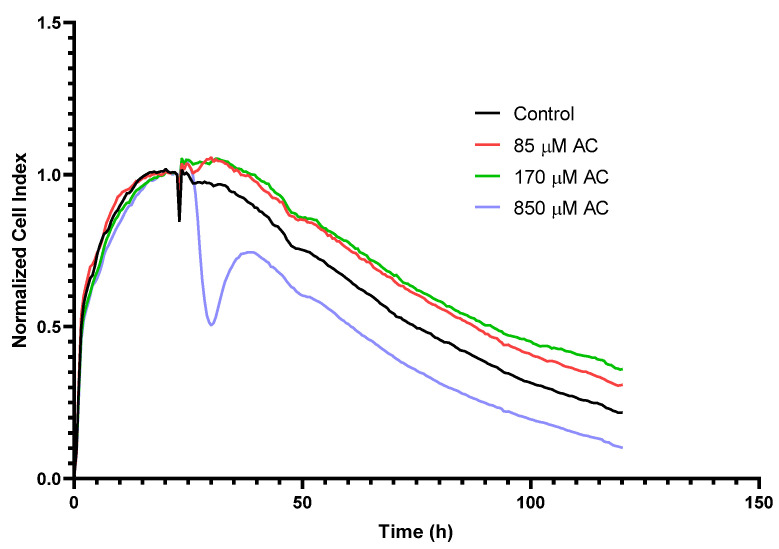
Impedance measurements with real time cell analyzer (RTCA), XCelligence system. Kinetics of epithelial cell response to treatment was monitored using cell index (CI). At the beginning of the plateau phase of growth, cells were treated with AC, at 85; 170; 850 μM concentrations. The Cis were normalized at the time of AC treatment.

**Figure 2 ijms-23-09036-f002:**
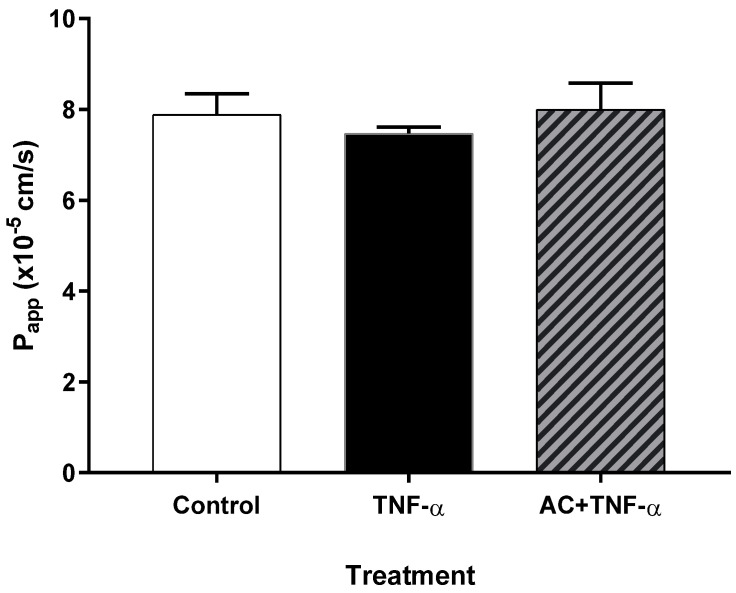
Apparent permeability coefficients (Papp) on HUVEC. Human umbilical vein endothelial cells (HUVECs) were pre-treated with TNF-α (10 ng/mL) and 85 μM AC + TNF-α (10 ng/mL) 24 h before permeability assay was performed. Permeability values were not significantly different; analysed by ANOVA.

**Figure 3 ijms-23-09036-f003:**
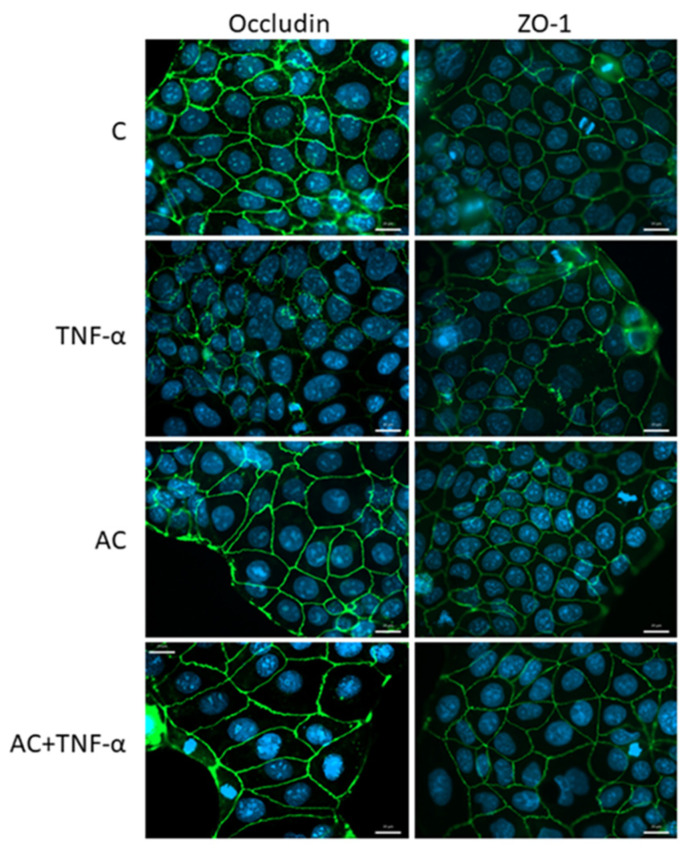
Effects of Anthocyanin (AC) and/or Tumor necrosis factor-α (TNF-α) treatment on junctional morphology of Caco-2 cells. Cells were treated for 24 h with culture medium (C), 50 ng/mL TNF-α, 100 µM AC or in combination of 100 µM AC (pretreatment) and 50 ng/mL TNF-α (AC + TNF-α). Immunostaining for Occludin and zonula occludens-1 (ZO-1) junctional proteins. Green color: immunostaining for Occludin, ZO-1. Blue color: staining of cell nuclei. Bar = 20 µm.

**Figure 4 ijms-23-09036-f004:**
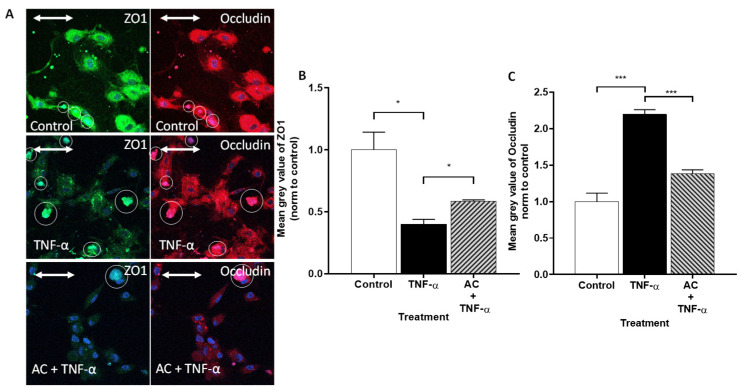
The effects of anthocyanin sour cherry extract on TNF-α induced ZO-1 and Occludin protein level. Human umbilical vein endothelial cells (HUVECs) were treated with TNF-α (10 ng/mL) and 85 μM AC + TNF-α (10 ng/mL) for 24 h. The protein level of ZO-1 and Occludin (**A**) were determined by immunostaining for Occludin and zonula occludens-1 (ZO-1) junctional proteins. Green color: immunostaining for ZO-1. Red color: immunostaining for Occludin. Blue color: staining of cell nucleus. Bar = 20 µm. The relative protein level of ZO-1 (**B**) and Occludin (**C**) were determined by mean gray value from immunostaining pictures. The means were normalized to untreated control. Data are presented as mean ± SEMs. *n* = 3. The significance of differences between sets of data was calculated by ANOVA. * *p* < 0.05, *** *p* < 0.001. The white circles indicate the regions that were not taken into account due to dye aggregation.

**Figure 5 ijms-23-09036-f005:**
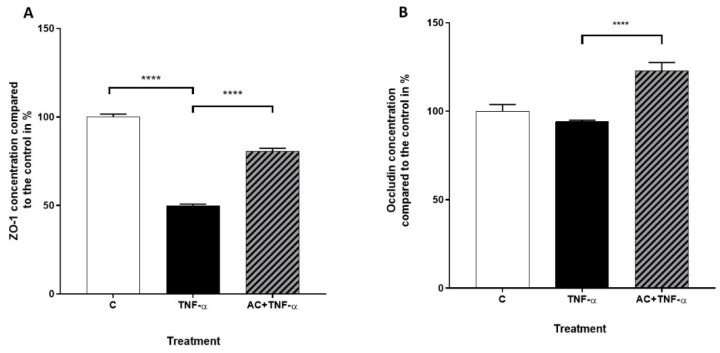
The effects of anthocyanin sour cherry extract on TNF-α induced ZO-1 and Occludin protein levels. Caco-2 were treated with TNF-α (50 ng/mL) and 100 µM AC + TNF-α (50 ng/mL) for 24 h. The protein levels of ZO-1 (**A**) and Occludin (**B**) were determined by Flow Cytometry and the means were normalized to un-treated control. Data are presented as mean ± SEMs. *n* = 3. The significance of differences between sets of data was calculated by ANOVA. **** *p* < 0.0001.

**Figure 6 ijms-23-09036-f006:**
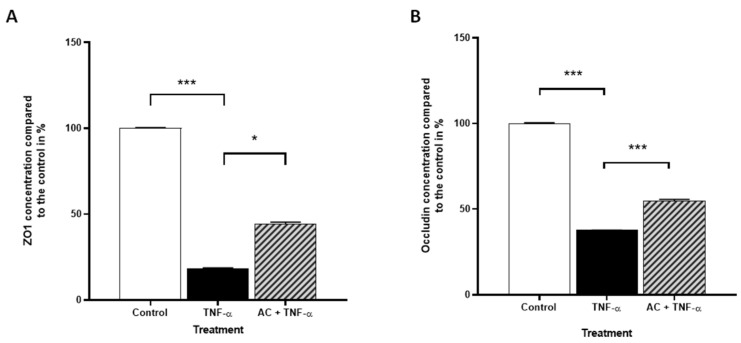
The effects of anthocyanin sour cherry extract on TNF-α induced ZO-1 and Occludin protein level. HUVECs were treated with TNF-α (10 ng/mL) and 85 µM AC + TNF-α (10 ng/mL) for 24 h. The protein level of ZO-1 (**A**) and Occludin (**B**) were determined by Flow Cytometry and the means were normalized to untreated control. Data are presented as mean ± SEMs. *n* = 3. The significance of differences between sets of data was calculated by ANOVA. * *p* < 0.05, *** *p* < 0.001.

**Figure 7 ijms-23-09036-f007:**
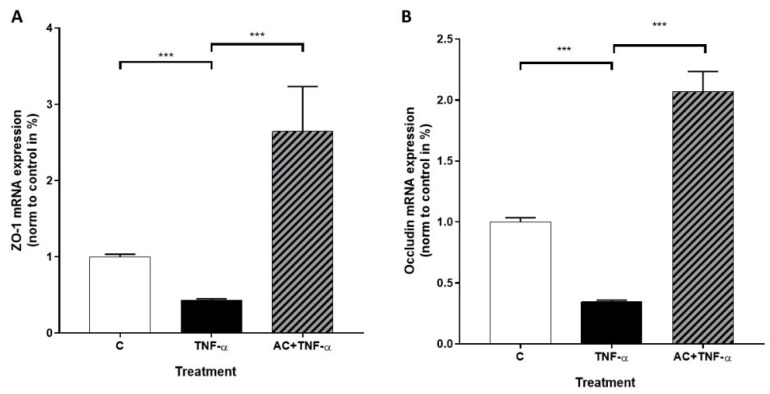
The effects of anthocyanin sour cherry extract on TNF-α induced ZO-1 and Occludin mRNA expression on Caco-2 cells. Cell cultures were treated with TNF-α (50 ng/mL) and 100 µM AC + TNF-α (50 ng/mL) for 24 h. The mRNA level of ZO-1 (**A**) and Occludin (**B**) were determined by RT-qPCR and target genes were normalized to GAPDH and control. Data are presented as mean ± SDs. *n* = 3. The significance of differences between sets of data was calculated by ANOVA. *** *p* < 0.001.

**Figure 8 ijms-23-09036-f008:**
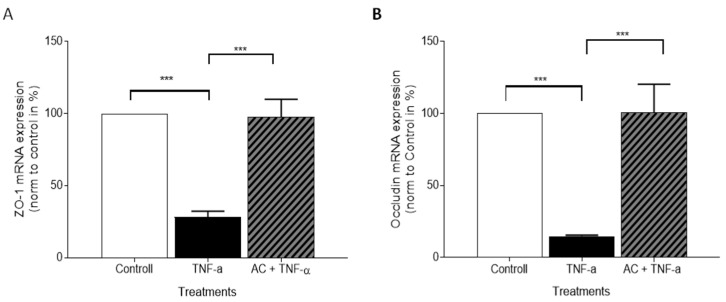
The effects of anthocyanin sour cherry extract on TNF-α induced ZO-1 and Occludin mRNA expression on HUVECs. Cell cultures were treated with TNF-α (10 ng/mL) and 85 μM AC + TNF-α (10 ng/mL) for 24 h. The mRNA level of ZO-1 (**A**) and Occludin (**B**) were determined by RT-qPCR and target genes were normalized to GAPDH and Control. Data are presented as mean ± SDs. *n* = 3. The significance of differences between sets of data was calculated by ANOVA. *** *p* < 0.001.

## Data Availability

Not applicable.

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
