# Peer review of "Comparison of the Modulating Effect of Anthocyanin-Rich Sour Cherry Extract on Occludin and ZO-1 on Caco-2 and HUVEC Cultures"

_ijms, 2022, doi:10.3390/ijms23169036_

Round 1

Reviewer 1 Report

Authors present data in which sour cherry anthocyanins modulate tight junction proteins in HUVECs and Caco-2 cells. While the paper is of interest, there are several clarifications and improvements needed prior to publication.

Minor comments:

Spell out abbreviations upon forst mention (e.g. ZO-1, TEER, etc.)

Please indicate lot number and sex of donor of HUVECs

Were glass slides pre-treated in any way to facilitate adherence of cells? Indicate as such.

Commas are used instead of periods lines 371 & 373

Standard deviation symbol used a few times starting line 384, not clear why.

Major comments

English language editing is needed

Since authors were working with a plant extract and state molar concentrations, how was molecular weight determined?

Regarding Figure 4, staining and respect quantification is highly unconvincing. Visually, AC expresses less ZO1 than TNF alone, and occludin is expressed less in AC vs control. Better images must be used or re-quantification is needed.

Author Response

Dear Reviewer!

At first, we would like to thank you for your comments, suggestions and valuable advices. We have revised our manuscript and we made corrections according to your recommendations. Our answers to your comments and suggestions are as follows:

Reviewer 2 Report

The paper is very interesting and well written. My small concern ralates to the impedance measurement with Real Time Cell Analyser. Why do we see a huge decrease in cells' number after 850uM AC? Is it a methodological mistake or was it done in one repetition? Also, section 2.3 is named immunohistochemistry, maybe renaming it to immunofluorescence would be better? It should be addressed. Aside from that the paper is interesting.

Author Response

Dear Reviewer!

Thank you for your comments and helpful advices. We have revised our manuscript according to your instruction. Our answers to your comments and suggestions are as follows.

Round 2

Reviewer 1 Report

Authors have addressed most comments, however, figures 3 & 4 are still of great concern. For transparency of findings, please include raw TIF files as a supplement (to be quantified by others if called into question) for the ICC images in figure 3 and figure 4. Additionally, it is needed to have arrows pointing to quantification sites that avoid non-specific staining since authors claim that non-specific binding contributes to the discrepancy between visual representation and quantification.

Author Response

Thanks for the comment. As you suggested, we marked the regions that were not taken into account during the evaluation, and also uploaded the microscopic images in a TIFF file.

Round 3

Reviewer 1 Report

Authors have adequately addressed all comments/concerns.